# Analysis of Risk Factors for High-Risk Lymph Node Metastasis in Papillary Thyroid Microcarcinoma

**DOI:** 10.3390/cancers17152585

**Published:** 2025-08-06

**Authors:** Yi-Hsiang Chiu, Shu-Ting Wu, Yung-Nien Chen, Wen-Chieh Chen, Lay-San Lim, Yvonne Ee Wern Chiew, Ping-Chen Kuo, Ya-Chen Yang, Shun-Yu Chi, Chen-Kai Chou

**Affiliations:** 1Division of Endocrinology and Metabolism, Department of Internal Medicine, Kaohsiung Chang Gung Memorial Hospital, Chang Gung University College of Medicine, 123 Ta-Pei Road, Niao-Sung District, Kaohsiung City 83301, Taiwan; a1080060@cgmh.org.tw (Y.-H.C.); b9602013@cgmh.org.tw (S.-T.W.); b9502055@cgmh.org.tw (Y.-N.C.); chingjing@cgmh.org.tw (W.-C.C.); laysan89@cgmh.org.tw (L.-S.L.); b9702027@cgmh.org.tw (Y.E.W.C.); ss830108@cgmh.org.tw (Y.-C.Y.); 2Division of General Surgery, Department of Surgery, Kaohsiung Chang Gung Memorial Hospital, Chang Gung University College of Medicine, 123 Ta-Pei Road, Niao-Sung District, Kaohsiung City 83301, Taiwan

**Keywords:** lymphatic metastasis, prognosis, thyroid neoplasms

## Abstract

Papillary thyroid microcarcinoma is a small thyroid cancer that generally has favorable outcomes, but some patients may still develop lymph node metastasis and disease recurrence. This study aimed to identify key factors associated with more aggressive disease, particularly high-risk lymph node metastasis. We found that male sex and extranodal extension were significantly associated with high-risk features. These findings may help clinicians better assess individual patient risk and formulate more personalized treatment strategies to improve long-term outcomes.

## 1. Introduction

According to the fifth edition of the World Health Organization Classification of Endocrine and Neuroendocrine Tumors, papillary thyroid microcarcinoma (PTMC) is defined as a tumor with a maximum diameter of ≤1 cm [1]. PTMC is generally characterized by an indolent clinical course compared to other forms of differentiated thyroid cancer [2,3,4]. Research indicates that PTMC has a 10-year disease-specific survival rate exceeding 98% [5,6], making it one of the least aggressive thyroid malignancies.

Despite the small size of the primary lesion, PTMC is associated with a relatively high incidence of lymph node metastasis (LNM), with reported rates ranging from 23% to 64% [7,8]. Several clinicopathological factors influence the propensity for LNM, including male sex, younger age (≤45 years), multifocal lesions, extrathyroidal extension (ETE), and a primary tumor size >6 mm, all of which have been identified as significant risk factors for central neck LNM in patients with PTMC [9,10]. These findings underscore the importance of thorough preoperative evaluation and tailored surgical strategies to address the metastatic potential of PTMC [11,12].

Predicting disease progression through nodal metastasis remains challenging due to the low sensitivity for detecting metastases in central lymph nodes. Certain pathological features are known to significantly increase the risk of local recurrence. Notably, the presence of high-volume LNM, ETE, and lateral neck LNM have been identified as key prognostic factors [4,13,14]. These findings suggest that certain subsets of patients with PTMC could benefit from aggressive therapeutic approaches.

This study, conducted at Kaohsiung Chang Gung Memorial Hospital, Taiwan, aimed to examine the clinicopathological features and outcomes of PTMC patients, and to determine the association between high-risk lymph node metastasis (HRLNM) and PTMC. For the purpose of this study, HRLNM is defined as high-volume LNM (>5 metastatic lymph nodes) and/or lateral neck LNM. Our findings aim to provide valuable insights for optimizing treatment strategies and reducing recurrence and distant metastasis rates.

## 2. Materials and Methods

### 2.1. Patient Recruitment

We conducted a retrospective review of the medical records of patients diagnosed with only PTMC who underwent thyroidectomy at the Kaohsiung Chang Gung Memorial Hospital, Taiwan, from 2013 to 2022. A clinical data system was used to identify eligible cases for the analysis. Of the 1201 patients, 111 were excluded based on one or more of the following criteria: a final pathology report indicating a maximum tumor diameter >1 cm, prior thyroid cancer surgery performed at another medical facility resulting in insufficient detailed pathological information, a diagnosis of double primary thyroid cancer, or an age <18 years. After applying these exclusion criteria, 1090 patients with only PTMC were identified. Additional exclusions were made for patients with a follow-up period of less than six months or missing initial thyroglobulin (Tg) levels postoperatively or during follow-up. Consequently, 985 patients (185 males, 800 females) were included in the final analysis, as illustrated in Figure 1.

Of these, 100 patients were found to have LNM, which specifically included central, lateral, or both central and lateral neck involvement, and were subsequently included in further investigations. They were then stratified for comparative analysis into groups based on the presence or absence of HRLNM. This study was approved by the Institutional Review Board of the Chang Gung Medical Foundation. (No. 202401692B0).

### 2.2. Treatment Protocol

Patients with PTMC undergoing the same management were monitored postoperatively according to our center’s protocol [15]. Detailed descriptions can be found in Appendix A.

### 2.3. Statistical Analysis

Comparisons of continuous variables were conducted using Student’s t-test or Wil-coxon test, while categorical variables were analyzed using the chi-square test or Fisher’s exact test, as appropriate. Recurrence-free survival (RFS) rates were estimated using Kaplan–Meier curves, and regression analyses of survival outcomes were performed using the Cox proportional hazards model. Statistical analyses were conducted using the Statistical Package for the Social Sciences (SPSS) software (version 25.0; IBM, Armonk, NY, USA). Statistical significance was set at *p* < 0.05.

## 3. Results

### 3.1. Patient Characteristics by Lymph Node Status

The clinical characteristics of the 100 patients (10.2%) with LNM compared to those without LNM are summarized in Table 1. The LNM group exhibited a lower mean age (47.6 ± 12.9 vs. 53.3 ± 11.7 years, *p* < 0.001) and a higher proportion of males (33.0% vs. 16.7%, *p* < 0.001). Post-thyroidectomy, the LNM group demonstrated more advanced AJCC-TNM staging (23.0% vs. 2.1%, *p* < 0.001), larger mean tumor size (0.6 ± 0.3 vs. 0.5 ± 0.3 cm, *p* < 0.001), a higher prevalence of lymph-vascular invasion (17.0% vs. 1.2%, *p* < 0.001), and a greater incidence of ETE (22.0% vs. 6%, *p* < 0.001). Sub-group analysis revealed a significantly higher proportion of V-Raf murine sarcoma viral oncogene homolog B1 (BRAF) mutations in the LNM group than in the non-LNM (75.6% vs. 58.6%; *p* = 0.029). After a median follow-up period of 4.5 years (interquartile range: 3.1–7.5 years), patients with LNM had a lower excellent treatment response rate (75% vs. 87%, *p* = 0.001) and a higher recurrence rate (9% vs. 0.6%, *p* = 0.001). Additionally, distant metastases occurred exclusively in the LNM group (two cases, 2%), with both patients presenting with multiple lung involvement.

### 3.2. Patient Characteristics by Anatomical Location of the Lymph Node Metastases

In Table 2, the clinicopathological characteristics of patients with LNM were further analyzed according to the involved compartment: central (*n* = 79), lateral (*n* = 9), and both central and lateral neck (*n* = 12). The median number of metastatic lymph nodes increased across the groups, with a median of 1 in the central group, 3 in the lateral group, and 6.5 in the both-compartment groups. The proportion of macrometastases was significantly higher in the both-compartment group (83.3%) compared to the central group (38.0%, *p* = 0.009). Female predominance was notably lower in the lateral group (22.2%) than in the central and both-compartment groups (72.2% and 66.7%, respectively; *p* = 0.011). Extranodal extension (ENE) was more frequent in patients with both-compartment involvement (58.3%) than in those with central LNM (15.4%, *p* = 0.006). However, no significant differences were observed between groups in age, tumor size, BRAF mutation status, recurrence, or treatment response.

### 3.3. Patient Characteristics by High-Risk Lymph Metastasis

The clinicopathological characteristics and outcomes of patients with and without HRLNM in PTMC are presented in Table 3. Among the 100 patients, 27 (27.0%) were categorized as having HRLNM. The mean age at diagnosis was similar between the two groups (47.48 ± 11.59 years for HRLNM vs. 47.68 ± 13.46 years for non-HRLNM, *p* = 0.947). The proportion of patients aged <55 years at the time of diagnosis did not differ significantly between the groups (70.4% vs. 64.4%, *p* = 0.575). The sex distribution differed between the groups, with a higher percentage of females in the non-HRLNM group (51.9% vs. 72.6%, *p* = 0.05). Regarding surgical management, most patients underwent total thyroidectomy with no significant difference between the groups (96.3% vs. 94.5%, *p* = 0.718). Tumor size was comparable between the groups (0.6 ± 0.2 cm for HRLNM vs. 0.6 ± 0.3 cm for non-HRLNM, *p* = 0.374). A higher percentage of patients with HRLNM had ETE (37.0% vs. 16.4%, *p* = 0.027) and ENE (37.0% vs. 15.3%, *p* = 0.018). No significant differences were observed in other pathological features such as tumor multiplicity, lympho-vascular invasion, or BRAF mutation status (*p* > 0.05). In terms of outcomes, there was no significant difference in the distribution of the TNM stage (AJCC-8th edition) between the groups (*p* = 0.135). The postoperative thyroglobulin levels were similar between the groups (*p* = 0.436). There was a significant difference in the local recurrence rate between the two groups (18.5% vs. 5.5%, *p* = 0.043). A higher proportion of patients with HRLNM received radioactive iodine therapy (96.3% vs. 82.2%, *p* = 0.104), had distant metastasis (3.7% vs. 1.4%, *p* = 0.469), and experienced overall mortality (3.7% vs. 0%, *p* = 0.27). Both groups had similar treatment responses, with 70.4% of patients with HRLNM achieving an excellent treatment response compared to 76.7% in the non-HRLNM group (*p* = 0.516).

### 3.4. Factors Associated with High-Risk Lymph Metastasis

Among the variables analyzed in Table 4, male sex was significantly associated with HRLNM in PTMC, with an odds ratio (OR) of 3.605 (95% confidence interval [CI] 1.06–12.25, *p* = 0.04). Tumor size (OR 0.34, 95% CI 0.04–2.77, *p* = 0.313), ETE (OR 2.162, 95% CI 0.57–8.16, *p* = 0.255), and BRAF mutation (OR 0.322, 95% CI 0.06–1.89, *p* = 0.21) were not significantly associated with HRLNM. Although age <55 years at diagnosis (OR 2.871, 95% CI 0.82–10.11, *p* = 0.1) and lymphocytic thyroiditis (OR 3.827, 95% CI 0.89–16.42, *p* = 0.071) demonstrated trends toward significance, these associations did not reach statistical significance. Furthermore, ENE was significantly associated with HRLNM (OR 3.764, 95% CI 1.04–13.6, *p* = 0.043). These findings suggest that the male sex and ENE are independently associated with an increased risk of HRLNM in patients with PTMC.

### 3.5. Recurrence-Free Survival Based on Lymph Node Metastasis and High-Risk Lymph Node Metastasis

RFS was significantly shorter in patients with LNM than in those without, with a mean duration of 120.9 vs. 198.6 months (*p* < 0.001; Figure 2). Similarly, patients with HRLNM exhibited a trend toward lower RFS (mean RFS: 113.5 months) compared to those without HRLNM (mean RFS: 124.6 months), with a *p*-value of 0.177. Although this difference was not statistically significant, the data suggest a potential association between HRLNM and decreased RFS, as shown in Figure 3.

## 4. Discussion

Beyond low-risk PTMC populations, both the presence and extent of LNM are strong predictors of unfavorable prognosis [16,17,18]. In this study, we defined HRLNM as either lateral neck involvement or high-volume nodal disease, representing patterns most closely linked to local recurrence and aggressive tumor behavior. Our data showed that patients with lateral neck LNM exhibited significantly higher numbers of metastatic lymph nodes and a greater proportion of macrometastases and extranodal extension, compared to those with central LNM. These characteristics suggest not only a higher disease burden but also distinct biological behavior. Lateral compartment metastases generally indicate more advanced lymphatic spread, and when coupled with extensive nodal volume, they may reflect a transition from localized to regionally aggressive disease [19,20]. Importantly, even though the patient numbers in the lateral LNM and both-compartment groups were smaller, the disproportionately higher rates of aggressive features imply that nodal distribution and volume could serve as practical surrogates for underlying tumor aggressiveness.

Multivariate analysis further identified ENE (OR: 3.76, *p* = 0.043) and male sex (OR: 3.61, *p* = 0.04) as independent risk factors for HRLNM in PTMC. ENE, defined as the spread of cancer cells beyond the lymph node capsule into surrounding tissue, reflects more aggressive tumor behavior and is associated with worse prognosis. Previous studies report that ENE increases the risk of recurrence and reduces survival (with a 10.3-fold increased recurrence rate compared to those without ENE) [21,22]. The observed link between male sex and more aggressive disease has been hypothetically attributed to the regulatory role of estrogen in thyroid cell proliferation [23,24]. However, current evidence supporting this association remains limited and requires validation in larger studies. A limitation of our study was the initial gender stratification, where males comprised only 18.7% of the cohort. This imbalance may have reduced the statistical significance of the finding regarding male sex. Collectively, these results underscore the importance of nodal distribution, nodal volume, and extranodal extension (ENE) in risk stratification. Furthermore, they indicate that male sex may represent an additional factor warranting further investigation.

Active surveillance (AS) has emerged as a viable management strategy for low-risk PTMC (T1aN0M0), demonstrating favorable long-term outcomes [25]. In our study, patients without lymph node metastasis (LNM) achieved an excellent treatment response rate of 87% and a very low recurrence rate of 0.6%, suggesting that conservative management may be appropriate in selected cases. These findings align with an increasing body of evidence indicating that immediate surgery may not be necessary for all patients with low-risk PTMC, provided adequate surveillance is maintained [26]. Successful implementation of AS depends on accurate diagnosis, appropriate risk stratification, informed patient consent, and structured long-term monitoring to ensure oncologic safety and timely intervention if necessary [27,28].

The prognostic significance of histological subtypes in PTC (papillary thyroid carcinoma)/PTMC has been increasingly emphasized in the fifth edition of the WHO Classification of Endocrine and Neuroendocrine Tumors, particularly for aggressive subtypes such as tall cell, hobnail, and columnar cell [29,30]. At our institution, systematic documentation of histological subtypes in PTMC began in 2022. As a result, many pathology reports from earlier years lacked subtype information. Among the cases with available data, aggressive subtype was rare and was not associated with adverse outcomes during a 7.3-year follow-up period. A summary of the documentation status and subtype distribution is provided in Appendix B and Figure A1.

Advancements in diagnostic technologies have significantly enhanced the detection of PTMC. Notably, its prevalence has risen, now accounting for approximately 30% of all PTC cases [31,32]. In our study, the initial LNM prevalence was 10.2%, which is considerably lower than previously reported rates. Several factors may explain this discrepancy. First, differences in surgical and pathological protocols could have contributed. At our institution, central neck dissection was selectively performed in cases with clinically evident LNM or other high-risk features, potentially leading to underestimation of occult metastases. In contrast, studies reporting higher LNM rates often employed routine prophylactic neck dissection or more extensive histopathological examinations, identifying more subclinical disease [8]. Second, our study cohort was derived from a single medical center in southern Taiwan, where regular health checkups and early detection programs are common. This may have increased the detection of early-stage PTMC before significant nodal involvement. Third, ethnic and regional differences may influence PTMC behavior. For example, one study showed that race and region of residence affect the risk of cervical LNM, with Black patients having a lower risk [33].

This study presents a large, well-characterized PTMC-only cohort with a median follow-up duration of nearly 10 years, providing valuable insight into clinical outcomes and risk stratification. However, certain limitations should be acknowledged. The single-center design may limit generalizability, and the relatively small number of LNM cases may limit statistical power. Molecular data such as BRAF mutation status were incomplete, as routine testing was only introduced in recent years. These limitations, along with regional and methodological variability in clinical protocols, may influence LNM detection rates. Future multicenter studies employing standardized protocols, extended follow-up durations, and comprehensive molecular profiling are warranted to validate these findings and further refine prognostic models in PTMC.

## 5. Conclusions

In this study of PTMC, male sex and ENE were independently associated with HRLNM. These features were linked to a higher risk of recurrence and a trend toward reduced RFS. Our findings highlight the importance of early risk stratification to guide surgical decision-making and postoperative follow-up. Tailored management strategies may improve clinical outcomes while minimizing overtreatment in patients with low-risk disease.

## Figures and Tables

**Figure 1 cancers-17-02585-f001:**
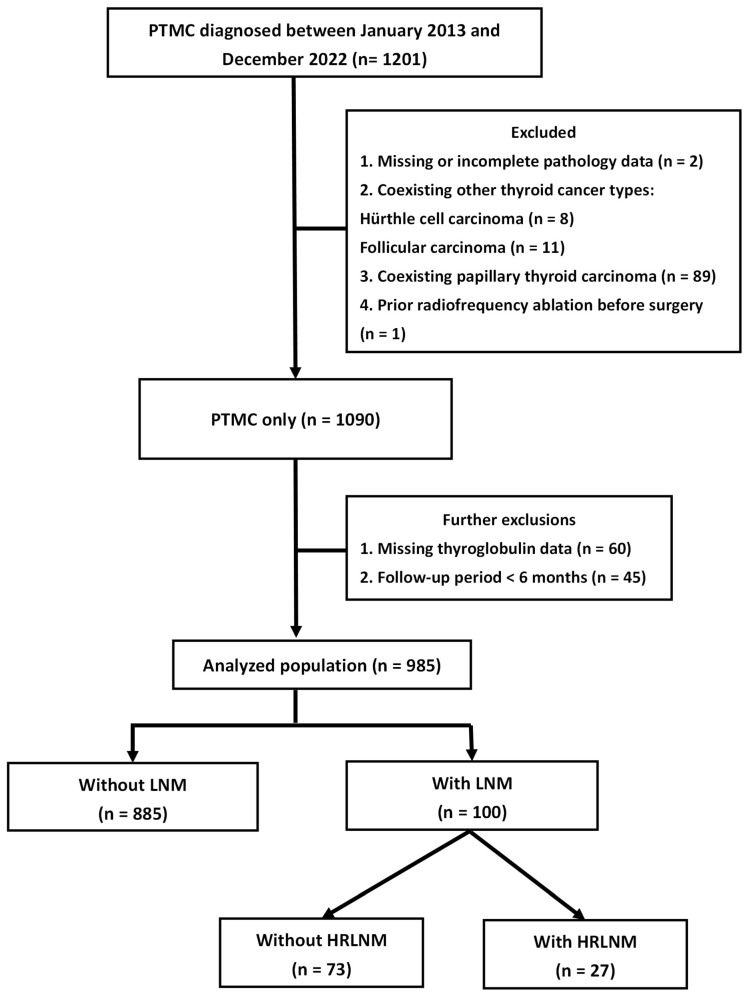
Algorithm for subject enrollment. Abbreviations: PTMC, papillary thyroid microcarcinoma; LN, lymph node metastasis; HRLNM, high-risk lymph node metastasis.

**Figure 2 cancers-17-02585-f002:**
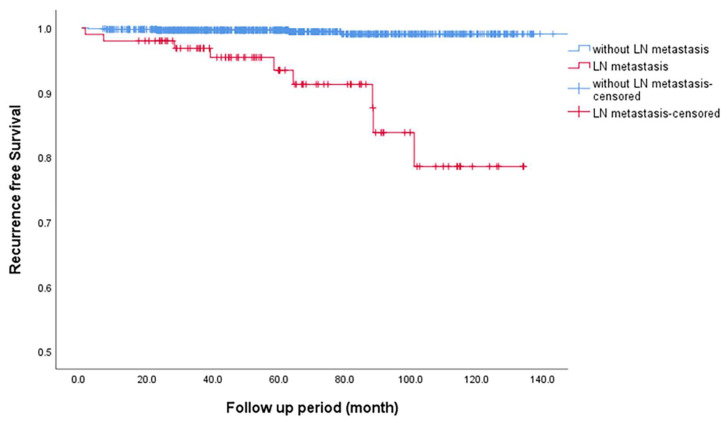
Kaplan–Meier recurrence-free survival curves for patients with and without lymph node metastasis. Abbreviation: LN, lymph node.

**Figure 3 cancers-17-02585-f003:**
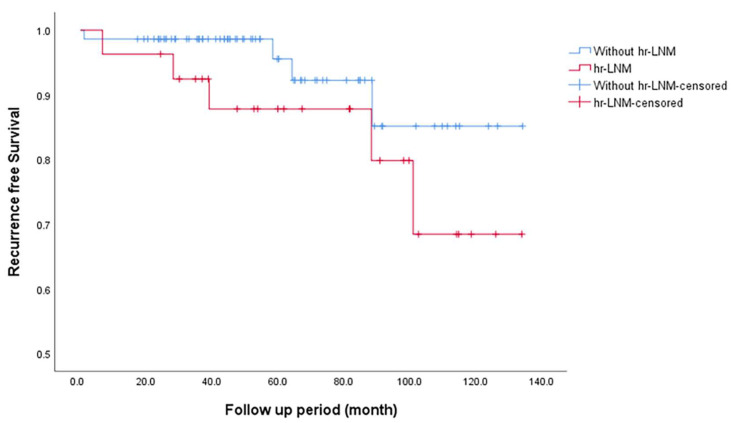
Kaplan–Meier recurrence-free survival curves for patients with and without high-risk lymph node metastasis. Abbreviation: hr-LNM, high-risk lymph node metastasis.

**Table 1 cancers-17-02585-t001:** Comparison of clinicopathologic characteristics with and without nodal metastasis; Abbreviations: LN, lymph node; BRAF, V-raf murine sarcoma viral oncogene homolog B1; TNM, tumor node metastasis; IQR, interquartile range; RAI, radioactive iodine. * BRAF mutation status was available in a subset of patients due to missing data.

Clinical Characteristic	LN Metastasis	Without LN Metastasis	*p* Value
Patient number (%)	100 (10.2)	885 (89.8)	
Mean age at diagnosis (y)	47.62 ± 12.93	53.33 ± 11.75	<0.001
Age < 55 years at diagnosis (%)	66 (66.0)	448 (50.6)	0.004
Female (%)	67 (67.0)	737 (83.3)	<0.001
Total thyroidectomy (%)	95 (95.0)	688 (77.7)	<0.001
Pathological			
Mean tumor size (cm)	0.6 ± 0.3	0.5 ± 0.3	<0.001
R2 resection (%)	17 (17.0)	26 (2.9)	<0.001
Lympho-vascular invasion (%)	17 (17.0)	11 (1.2)	<0.001
Extrathyroidal extension (%)	22 (22.0)	53 (6.0)	<0.001
Multiplicity (%)	46 (46.0)	301 (34.0)	0.017
Lymphocytic thyroiditis (%)	16 (16.0)	151 (17.1)	0.785
BRAF mutation (%) *	34 (75.6)	181 (58.6)	0.029
Outcome			
TNM stage (AJCC-8th edition)			<0.001
Stage I + II (%)	77 (77.0)	866 (97.9)	
Stage III + IV (%)	23 (23.0)	19 (2.1)	
Median follow up period after diagnosis (yr)(IQR)	4.5 (3.1–7.5)	5.1 (3.2–7.4)	0.847
Postoperative thyroglobulin (IQR)	0.49 (0.2–3.72)	0.5 (0.2–5.73)	0.26
Received RAI(%)	86 (86.0)	90 (10.2)	<0.001
Recurrence (%)	9 (9.0)	5 (0.6)	0.001
Distant metastasis (%)	2 (2.0)	0	0.01
Overall mortality (%)	1 (1.0)	5 (0.6)	0.475
Treatment response, excellent (%)	75 (75.0)	770 (87.0)	0.001

**Table 2 cancers-17-02585-t002:** Comparison of clinicopathological characteristics according to the location of lymph node metastases. Abbreviations: BRAF, V-raf murine sarcoma viral oncogene homolog B1; TNM, tumor node metastasis; IQR, interquartile range; RAI, radioactive iodine. † *p* < 0.05 vs. central in the Bonferroni post hoc; ‡ *p* < 0.05 vs. lateral in the Bonferroni post hoc; * BRAF mutation status was available in a subset of patients due to missing data.

Location	Central	Lateral	Both	*p* Value
Patient number	79	9	12	
Mean age at diagnosis (y)	46.91 ± 13.52	53.33 ± 9.71	48 ± 10.39	0.371
Age < 55 years at diagnosis (%)	53 (67.1)	5 (55.6)	8 (66.7)	0.749
Female (%)	57 (72.2)	2 (22.2) †	8 (66.7)	0.011
Total thyroidectomy (%)	75 (94.9)	8 (88.9)	12 (100)	0.461
Pathological				
Number of metastatic lymph nodes (IQR)	1 (1–3)	3 (2–4) †	6.5(3–12) †‡	<0.001
Macrometastases (%)	30 (38)	5 (55.6)	10 (83.3) †	0.009
Mean tumor size (cm)	0.7 ± 0.2	0.5 ± 0.3	0.6 ± 0.2	0.125
R2 resection (%)	12 (15.2)	2 (22.2)	3 (25)	0.548
Lympho-vascular invasion (%)	11 (13.9)	3 (33.3)	3 (25)	0.178
Extrathyroidal extension (%)	15 (19)	3 (33.3)	4 (33.3)	0.334
Multiplicity (%)	35 (44.3)	4 (44.4)	7 (58.3)	0.673
Extranodal extension (%)	12 (15.4)	2 (22.2)	7 (58.3) †	0.006
Lymphocytic thyroiditis (%)	13 (16.5)	1 (11.1)	2 (16.7)	1
BRAF mutation (%) *	29 (76.3)	2 (100)	3 (60)	0.767
Outcome				
TNM stage (AJCC-8th edition)				0.046
Stage I + II (%)	64 (81)	4 (44.4)	9 (75)	
Stage III + IV (%)	15 (19)	5 (55.6)	3 (25)	
Median follow up period after diagnosis (yr) (IQR)	4.3 (3.0–7.2)	8.1 (2.6–9.7)	5.3 (4.4–7.3)	0.355
Postoperative thyroglobulin (IQR)	0.50 (0.2–3.77)	8.18 (2.63–9.71)	5.38 (4.41–7.39)	0.998
Received RAI(%)	66 (83.5)	8 (88.9)	12 (100)	0.375
Recurrence (%)	7 (8.9)	1 (11.1)	1 (8.3)	1
Distant metastasis (%)	1 (1.3)	1 (11.1)	0	0.09
Overall mortality (%)	0	1 (11.1)	0	0.27
Treatment response, excellent (%)	59 (74.7)	7 (77.8)	9 (75)	1

**Table 3 cancers-17-02585-t003:** Comparison of clinicopathologic characteristics with and without high-risk lymph node metastasis. Abbreviations: HRLNM, high-risk lymph node metastasis; LN, lymph node; BRAF, V-raf murine sarcoma viral oncogene homolog B1; TNM, tumor node metastasis; IQR, interquartile range; RAI, radioactive iodine. * BRAF mutation status was available in a subset of patients due to missing data.

Characteristic	HRLNM	Non-HRLNM	*p* Value
Patient number (%)	27 (27.0)	73 (73.0)	
Mean age at diagnosis (y)	47.48 ± 11.59	47.68 ± 13.46	0.947
Age < 55 years at diagnosis (%)	19 (70.4)	47 (64.4)	0.575
Female (%)	14 (51.9)	53 (72.6)	0.05
Total thyroidectomy (%)	26 (96.3)	69 (94.5)	0.718
Pathological			
Mean tumor size (cm)	0.6 ± 0.2	0.6 ± 0.3	0.374
R2 resection (%)	7 (25.9)	10 (13.7)	0.228
Lympho-vascular invasion (%)	6 (22.2)	11 (15.1)	0.387
Extrathyroidal extension (%)	10 (37.0)	12 (16.4)	0.027
Multiplicity (%)	13 (48.1)	33 (45.2)	0.793
Extranodal extension (%)	10 (37.0)	11 (15.3)	0.018
Lymphocytic thyroiditis (%)	5 (18.5)	11 (15.1)	0.76
BRAF mutation (%) *	7 (77.8)	27 (75.0)	0.862
Outcome			
TNM stage (AJCC-8th edition)			0.135
Stage I + II (%)	18 (66.7)	59 (80.8)	
Stage III + IV (%)	9 (33.3)	14 (19.2)	
Median follow up period after diagnosis (yr) (IQR)	5.6 (3.3–8.6)	4.3 (3.0–7.1)	0.094
Postoperative thyroglobulin (IQR)	0.50 (0.20–2.35)	0.49 (0.20–3.78)	0.436
Received RAI	26 (96.3)	60 (82.2)	0.104
Recurrence (%)	5 (18.5)	4 (5.5)	0.043
Distant metastasis (%)	1 (3.7)	1 (1.4)	0.469
Overall mortality, *n* (%)	1 (3.7)	0	0.27
Treatment response, excellent (%)	19 (70.4)	56 (76.7)	0.516

**Table 4 cancers-17-02585-t004:** Multivariate logistic regression analysis of factors associated with high-risk lymph node metastasis. Abbreviation: BRAF, V-raf murine sarcoma viral oncogene homolog B1.

Variables	Hazard Ratio (95%CI)	*p* Value
Age:		
Age < 55 years at diagnosis	2.87 (0.82–10.11)	0.1
Sex:		
Female	Reference	
Male	3.61 (1.06–12.25)	0.04
Tumor size	0.34 (0.04–2.77)	0.313
Extrathyroid extension:		
Absent	Reference	
Present	2.16 (0.57–8.16)	0.255
Lymphocytic thyroiditis:		
Absent	Reference	
Present	3.83 (0.89–16.42)	0.071
Extranodal extension:		
Absent	Reference	
Present	3.76 (1.04–13.6)	0.043
BRAF mutation:		
Absent	Reference	
Present	0.32 (0.06–1.89)	0.21

## Data Availability

Data is unavailable due to privacy or ethical restrictions.

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
