# Peer review of "Analysis of Risk Factors for High-Risk Lymph Node Metastasis in Papillary Thyroid Microcarcinoma"

_cancers, 2025, doi:10.3390/cancers17152585_

Round 1
Reviewer 1 Report
Comments and Suggestions for Authors
topics in contemporary thyroid pathology, offering new insights through their study results.
Here are my suggestions:
Introduction
Lines 64–65: “This study aimed to examine the clinicopathological factors and outcomes of patients with PTMC in Taiwan.”
This statement appears overly generic. I recommend rephrasing it to better align with what is described in the abstract and methods section. The data were obtained from a single medical centre, not from a city-wide population, and this should be clearly stated.
Lines 66–67: Please correct the typographical error in the sentence: “We defined HRLNM is defined as high-volume LNM (>5 metastatic lymph nodes) and/or lateral neck LNM.”
Additionally, if the definition of HRLNM does not follow a specific guideline (e.g., the ATA risk stratification), I suggest making it clearer that this definition was proposed specifically for the purposes of your study. You might consider integrating this into the study aim itself. For example, lines 87–90 currently read: “Of these, 100 presented with LNM and were included in further investigations. HRLNM is defined as the presence of high-volume LNM (>5) and/or lateral neck LNM, which are mentioned in the American Thyroid Association (ATA) risk of structural disease recurrence [4].”
This concept would be better placed in the Introduction, where you define the study objective and scope. Moreover, the paragraph from lines 64–69 lacks fluency; I suggest reworking it for clarity and flow. The final sentence -“We aimed to provide valuable insights for optimizing treatment strategies and minimizing recurrence and distant metastasis rates.” - could be shortened, as it seems somewhat redundant.
Materials and Methods
Lines 72–83: You may consider adding the presence of non-papillary thyroid carcinoma (e.g., follicular, poorly differentiated, medullary, anaplastic) as part of the exclusion criteria. If only papillary thyroid microcarcinomas (PTMCs) were included, this should be stated clearly in both the text and Figure 1. In the figure, you mention that coexisting medullary and follicular carcinomas were excluded, please clarify that only micro-PTCs were analysed. (Also, did you include multifocal microcarcinomas?)
Line 87: “Of these, 100 presented with LNM and were included in further investigations.”
Please clarify whether patients with central and/or lateral lymph node metastases were included, particularly since the sentence following this will be moved to the Introduction (i.e., “HRLNM is defined as…”). This distinction should be clearly defined in this section.
Results
I recommend stratifying your cohort based on the anatomical location of the lymph node metastases: central, lateral (specifying the levels), or both. Also, please provide the number of metastatic lymph nodes per compartment, along with their size (i.e., micrometastases vs. macrometastases).
Discussion
The opening paragraph discusses active surveillance (lines 188–197), which may come across as somewhat disconnected from your findings. Since your results suggest that some PTMCs may behave aggressively, I believe your discussion should start by highlighting this message. Active surveillance can then be discussed later in the section, in contrast to the higher-risk subgroup you identified. I also encourage you to elaborate further on the analyses proposed above in the Results section, as doing so would enhance the impact of your findings.
Thank you. I hope these suggestions are helpful in strengthening your manuscript.
Author Response
Comments 1: [Lines 64–65: “This study aimed to examine the clinicopathological factors and outcomes of patients with PTMC in Taiwan.” This statement appears overly generic. I recommend rephrasing it to better align with what is described in the abstract and methods section. The data were obtained from a single medical centre, not from a city-wide population, and this should be clearly stated.]
Response 1: Thank you for pointing this out. We agree with this comment. Therefore, we have revised as follows [Page 2, lines 61-62: This study, conducted at Kaohsiung Chang Gung Memorial Hospital, Taiwan, aimed to examine the clinicopathological features and outcomes of PTMC patients,]
Comments 2: [Lines 66–67: Please correct the typographical error in the sentence: “We defined HRLNM is defined as high-volume LNM (>5 metastatic lymph nodes) and/or lateral neck LNM.”
Additionally, if the definition of HRLNM does not follow a specific guideline (e.g., the ATA risk stratification), I suggest making it clearer that this definition was proposed specifically for the purposes of your study. You might consider integrating this into the study aim itself. For example, lines 87–90 currently read: “Of these, 100 presented with LNM and were included in further investigations. HRLNM is defined as the presence of high-volume LNM (>5) and/or lateral neck LNM, which are mentioned in the American Thyroid Association (ATA) risk of structural disease recurrence [4].”
This concept would be better placed in the Introduction, where you define the study objective and scope. Moreover, the paragraph from lines 64–69 lacks fluency; I suggest reworking it for clarity and flow. The final sentence -“We aimed to provide valuable insights for optimizing treatment strategies and minimizing recurrence and distant metastasis rates.” - could be shortened, as it seems somewhat redundant.]
Response 2: We have corrected typographical errors in the manuscript. As suggested, we've now shifted the definition of HRLNM solely to the Introduction and removed its mention from the Methods section, clarifying that this definition is specific to our study. Furthermore, we've rewritten previous sentences on lines 64-69 to improve their fluency. [Page 2, line 61-66: This study, conducted at Kaohsiung Chang Gung Memorial Hospital, Taiwan, aimed to examine the clinicopathological features and outcomes of PTMC patients, and to determine the association between high-risk lymph node metastasis (HRLNM) and PTMC. For the purpose of this study, HRLNM is defined as high-volume LNM (>5 metastatic lymph nodes) and/or lateral neck LNM. Our findings aim to provide valuable insights for optimizing treatment strategies and reducing recurrence and distant metastasis rates.]
Comments 3: [Materials and Methods
Lines 72–83: You may consider adding the presence of non-papillary thyroid carcinoma (e.g., follicular, poorly differentiated, medullary, anaplastic) as part of the exclusion criteria. If only papillary thyroid microcarcinomas (PTMCs) were included, this should be stated clearly in both the text and Figure 1. In the figure, you mention that coexisting medullary and follicular carcinomas were excluded, please clarify that only micro-PTCs were analysed. (Also, did you include multifocal microcarcinomas?)]
Response 3: In the methods, we did indeed analyze only PTMC population. Multifocal PTMC was also included in the study. However, we performed a pre-screening to ensure that each foci met the criteria for PTMC. Following your suggestion for greater clarity, we have revised the text and figures as follows: [Page 2, lines 69-71: We conducted a retrospective review of the medical records of patients diagnosed with only PTMC who underwent thyroidectomy at the Kaohsiung Chang Gung Memorial Hospital, Taiwan, from 2013 to 2022. Page 2, line 76-77: After applying these exclusion criteria, 1,090 patients with only PTMC were identified. Figure 1: change the PTMC alone to PTMC only]
Comments 4: [Line 87: “Of these, 100 presented with LNM and were included in further investigations.”
Please clarify whether patients with central and/or lateral lymph node metastases were included, particularly since the sentence following this will be moved to the Introduction (i.e., “HRLNM is defined as…”). This distinction should be clearly defined in this section.]
Response 4: We agree with your suggestion and have made the following revisions: We've now more clearly defined the lymph node metastasis (LNM) included population, and rewritten subsequent sentences to improve their fluency. [Page 4, lines 84-87: Of these, 100 patients were found to have LNM, which specifically included central, lateral, or both central and lateral neck involvement, and were subsequently included in further investigations. They were then stratified for comparative analysis into groups based on the presence or absence of HRLNM.]
Comments 5: [Results
I recommend stratifying your cohort based on the anatomical location of the lymph node metastases: central, lateral (specifying the levels), or both. Also, please provide the number of metastatic lymph nodes per compartment, along with their size (i.e., micrometastases vs. macrometastases).]
Response 5:
We have added a new Table 2 to present the clinicopathological characteristics of patients stratified by the location of lymph node metastases. This detailed breakdown includes data on the number of metastatic lymph nodes, the proportion of macrometastases, and other relevant clinicopathological factors. [Page 5, lines 126-138: In table 2, the clinicopathological characteristics of patients with LNM were further analyzed according to the involved compartment—central (n = 79), lateral (n = 9), and both central and lateral neck (n = 12). The number of metastatic lymph nodes increased across the groups, with a median of 1 in the central group, 3 in the lateral group, and 6.5 in the both-compartment group. The proportion of macrometastases was significantly higher in the both-compartment group (83.3%) compared to the central group (38.0%, p = 0.009). Female predominance was notably lower in the lateral group (22.2%) than in the central and both-compartment groups (72.2% and 66.7%, respectively; p = 0.011). Extranodal extension was more frequent in patients with both-compartment involvement (58.3%) than in those with central LNM (15.4%, p = 0.006). However, no significant differences were observed between groups in age, tumor size, BRAF mutation status, recurrence, or treatment response.]
Comments 6: [Discussion
The opening paragraph discusses active surveillance (lines 188–197), which may come across as somewhat disconnected from your findings. Since your results suggest that some PTMCs may behave aggressively, I believe your discussion should start by highlighting this message. Active surveillance can then be discussed later in the section, in contrast to the higher-risk subgroup you identified. I also encourage you to elaborate further on the analyses proposed above in the Results section, as doing so would enhance the impact of your findings.]
Response 6: We appreciate your valuable feedback! Following your suggestion, we've revised the initial paragraph of the Discussion section. This now includes a more in-depth exploration of our findings, grounding the discussion directly in the presented results. Additionally, we've repositioned the active surveillance discussion to a subsequent paragraph. [Page 10-11, lines 207-243: Beyond low-risk PTMC populations, various clinicopathologic factors predict structural recurrence [16], including tumor diameters >5 mm, a central neck LNM ratio >0.5 [17], younger age (<45 years), male sex [18,19], multifocality [20], capsular invasion [21], and ETE [7]. Consistent with these risk factors, our data showed that patients with lateral neck LNM exhibited significantly higher numbers of metastatic lymph nodes and a greater proportion of macrometastases and extranodal extension, compared to those with central LNM. These characteristics suggest not only a higher disease burden but also distinct biological behavior. Lateral compartment metastases generally indicate more advanced lymphatic spread, and when coupled with extensive nodal volume, they may reflect a transition from localized to regionally aggressive disease [22,23]. Importantly, even though the patient numbers in the lateral LNM and both-compartment groups were smaller, the disproportionately higher rates of aggressive features imply that nodal distribution and volume could serve as practical surrogates for underlying tumor aggressiveness. Consequently, high-volume LNM and lateral neck LNM were identified as high-risk features in this study.
In our study, multivariate analysis identified ENE (OR: 3.76, p = 0.043) and male sex (OR: 3.61, p = 0.04) as independent risk factors for HRLNM in PTMC. The association between male sex and poorer prognosis in PTC has been linked to estrogen’s regulatory role in thyroid cell proliferation [24,25], potentially contributing to sex-based differences in tumor aggressiveness and outcomes. ENE, defined as the spread of cancer cells beyond the lymph node capsule into surrounding tissue, reflects more aggressive tumor behavior and is associated with worse prognosis. Previous studies report that ENE increases the risk of recurrence and reduces survival (with a 10.3-fold increased recurrence rate compared to those without ENE) [26,27]. These findings highlight the need to incorporate ENE into risk assessment and treatment planning for PTMC. Early identification of high-risk features can guide the extent of surgical intervention and inform long-term surveillance strategies.
Active surveillance (AS) has emerged as a viable management strategy for low-risk PTMC (T1aN0M0), demonstrating favorable long-term outcomes [28]. In our study, patients without lymph node metastasis (LNM) achieved an excellent treatment response rate of 87% and a very low recurrence rate of 0.6%, suggesting that conservative management may be appropriate in selected cases. These findings align with an increasing body of evidence indicating that immediate surgery may not be necessary for all patients with low-risk PTMC, provided adequate surveillance is maintained [29]. Successful implementation of AS depends on accurate diagnosis, appropriate risk stratification, informed patient consent, and structured long-term monitoring to ensure oncologic safety and timely intervention if necessary [30,31].]
Reviewer 2 Report
Comments and Suggestions for Authors
The article by Сhiu Y.-H. et al is a retrospective study of the medical records of patients with only papillary thyroid microcarcinoma (PTMC) as an initial diagnosis who underwent thyroidectomy at the same Memorial Hospital in South Taiwan. The aim of the authors was to elucidate risk factors associated with aggressive form of PTMC which generally has quite favorable outcomes but sometimes develops lymph node metastasis (LNM) and disease recurrence. The study is well organized, used adequate mathematical statistic analysis of data but has a number of limitations noted by the authors themselves. First of all, the number of patients in some of the analyzed groups is too small. For example, patients suffering from LNM make up only 100 men, slightly more than 10% of all participants with PTMC (985) and the group with hr-LNM only 27 from 100. As a result for quite objective reasons when the authors tried to compare clinicopathological characteristics associated with location of LNM (Table 2) the number of items in some analyzed groups decreased to 1,2,3 cases which is absolutely not
suitable for the analysis of statistical patterns. Data in Table 2 present analysis of multiple variable groups (central, lateral, both) according to Bonferroni post hoc, but from the explanations to the table it is not always possible to understand comparison of which groups corresponds to the values of mathematical reliability in the last column.
At last multivariate logistic regression analysis revealed extranodal extension (ENE)(OR 3.61,p=0.043) and male sex (OR 3.61 and p=0.04) as independent risk factors in development hr-LNM in PTMC. The authors consider estrogen’s regulatory role in proliferation of thyroid cells as a possible explanation of phenomenon. Unfortunately the initial gender stratification was not made correctly (there were only 185 males and 800 females as a participants) which reduces the significance of the result obtained. At the same time the significance of hr-LNM and LNM itself as risk factors for unfavorable prognosis in PTMC are quite convincing and make the study reasonable and fit for publication after small correction.
Thus, the article can be published after: 1. the authors provide more explanations for Table 2, 2. the emphasis in determining the main result obtained should be shifted towards the role of rh-LNM and LNM as main risk factors for unfavorable prognosis in PTMC as male sex should be mentioned as additional risk factor but future studies are necessary to confirm this preliminary observation, 3.in line 131 compartment group must be changed for groups.
Author Response
Comments 1: [Data in Table 2 present analysis of multiple variable groups (central, lateral, both) according to Bonferroni post hoc, but from the explanations to the table it is not always possible to understand comparison of which groups corresponds to the values of mathematical reliability in the last column. The authors provide more explanations for Table 2]
Response 1: We sincerely appreciate this observation. We acknowledge that some LNM location subgroups are small, which is an inherent limitation of our single‑center cohort. This analysis was included in response to another reviewer’s suggestion to compare clinicopathologic features by lymph node location. To address the small sample size, Fisher’s exact test was applied in SPSS to ensure statistical validity. We also recognize that our initial presentation of group comparisons in Table 2 was unclear due to a misinterpretation of the SPSS output; the footnote and symbol annotations have now been corrected accordingly. [Page 6, line 142: § p < 0.05 in post hoc comparison;† p < 0.05 vs. central in the Bonferroni post hoc; ‡ p < 0.05 vs. lateral in the Bonferroni post hoc].
Comments 2: [the emphasis in determining the main result obtained should be shifted towards the role of rh-LNM and LNM as main risk factors for unfavorable prognosis in PTMC as male sex should be mentioned as additional risk factor but future studies are necessary to confirm this preliminary observation]
Response 2: We appreciate your constructive suggestion. In response, we have revised the first and second paragraphs of the discussion to more clearly emphasize the relationship between HRLNM and clinical outcomes. Furthermore, while male sex was identified as an independent risk factor for HRLNM, we have explicitly noted that this finding requires further validation in future studies. [Page 10-11, line 207-235: Beyond low-risk PTMC populations, both the presence and extent of LNM are strong predictors of unfavorable prognosis [16-18]. In this study, we defined HRLNM as either lateral neck involvement or high-volume nodal disease, representing patterns most closely linked to local recurrence and aggressive tumor behavior. Our data showed that patients with lateral neck LNM exhibited significantly higher numbers of metastatic lymph nodes and a greater proportion of macrometastases and extranodal extension, compared to those with central LNM. These characteristics suggest not only a higher disease burden but also distinct biological behavior. Lateral compartment metastases generally indicate more advanced lymphatic spread, and when coupled with extensive nodal volume, they may reflect a transition from localized to regionally aggressive disease [19,20]. Importantly, even though the patient numbers in the lateral LNM and both-compartment groups were smaller, the disproportionately higher rates of aggressive features imply that nodal distribution and volume could serve as practical surrogates for underlying tumor aggressiveness.
Multivariate analysis further identified ENE (OR: 3.76, p = 0.043) and male sex (OR: 3.61, p = 0.04) as independent risk factors for HRLNM in PTMC. ENE, defined as the spread of cancer cells beyond the lymph node capsule into surrounding tissue, reflects more aggressive tumor behavior and is associated with worse prognosis. Previous studies report that ENE increases the risk of recurrence and reduces survival (with a 10.3-fold increased recurrence rate compared to those without ENE) [21,22]. The observed link between male sex and more aggressive disease has been hypothetically attributed to the regulatory role of estrogen in thyroid cell proliferation [23, 24]. However, current evidence supporting this association remains limited and requires validation in larger studies. A limitation of our study was the initial gender stratification, where males comprised only 18.7% of the cohort. This imbalance may have reduced the statistical significance of the finding regarding male sex. Collectively, these results underscore the importance of nodal distribution, nodal volume, and extranodal extension (ENE) in risk stratification. Furthermore, they indicate that male sex may represent an additional factor warranting further investigation.]
Comments 3: [in line 131 compartment group must be changed for groups.]
Response 3: We have corrected the wording by adding the missing plural form. [Page 5, line 131: , and 6.5 in the both-compartment groups.]
Round 2
Reviewer 1 Report
Comments and Suggestions for Authors
Thank you for addressing my comments and for clarifying the points previously raised. The revised manuscript is now much clearer and more effective in conveying its message.
Author Response
Thank you very much for your valuable feedback. We truly appreciate your time and effort in reviewing our manuscript.